# Variation in Obsessive-Compulsive Disorder Symptoms and Treatments: A Side Effect of COVID-19

**DOI:** 10.3390/ijerph18147420

**Published:** 2021-07-12

**Authors:** Wuqianhui Liu, Haitao Zhang, Yuan He

**Affiliations:** 1The First School of Clinical Medicine, Nanjing Medical University, Nanjing 211166, China; liuwqh@njmu.edu.cn; 2The Research Center for Medical Security, China Pharmaceutic University, Nanjing 211166, China; andy_zhang88@sina.com; 3The Institute of National Governance and National Audit, Nanjing Audit University, Nanjing 211815, China; 4The Institute of Medical Humanities, Nanjing Medical University, Nanjing 211166, China

**Keywords:** obsessive-compulsive disorder, COVID-19, treatment, symptom

## Abstract

The Coronavirus Disease 2019 (COVID-19) exerts variable impact on patients with obsessive compulsive disorders (OCD). There remains a challenge to determine the extent to which OCD is exacerbated due to the pandemic. Therefore, our aim is to explicate the latest researching progress of OCD under COVID-19 based on a review of 15 existing articles. Our review confirms the prevalence of OCD exacerbation in different age groups and particular symptoms. However, it also reveals nonconformity among research, lack of investigation in OCD treatment, and imbalance in OCD symptoms research. Further, we discuss the probable reasons of the exacerbation and current situation of OCD treatments. Finally, based on our discussion, we offer suggestions on how to manage OCD under the new circumstance, including the introduction of new policies, the use of communications technology, the improvement of researching methods, and possible angles for further research.

## 1. Introduction

The Coronavirus Disease 2019 (COVID-19) is a contagious respiratory disease which has greatly affected the world, and was defined as a pandemic on 11 March 2020 [1]. The world has recently reached an unpalatable milestone of 100 million patients and 2 million related deaths [2]. This fast-spread disease undoubtedly causes worldwide anxiety. Besides the potential probability of being infected, the general adaption of quarantine measures, the worrisome slowdown of economies, and the curtailment of forms of social interaction are all factors contributing to greater mental health concern during the pandemic. Studies investigating public mental health during COVID-19 are constantly progressing. A COVID Stress Syndrome, which is mainly characterized as being anxious and frightened to be contaminated, has been introduced [3]. Despite the fact that contamination fear is a type of OCD symptom, there is minor focus on patients with obsessive-compulsive disorder (OCD), the life prevalence of which is approximately 2% of the population. Patients’ vulnerability may be strengthened during COVID-19; exacerbation of OCD symptoms, especially contamination fear, has been reported following previous epidemic disease outbreaks such as SARS-CoV [4], partly since both diseases arouse a fear of infection, which is a trigger of contamination fear [5], and requires a thorough hand-washing [6]. Though effective studies have been done, the nonnegligible impact of COVID-19 on OCD is still not adequately investigated. OCD symptomatology and treatment trajectory post-pandemic need further research. In this context, we intend to explicate current explorative findings about variations in OCD under COVID-19, mainly focusing on symptoms, treatments, and related factors, based on existing literature. Our aim is to map the current situation of OCD in this special period of time, and provide researchers of the same or similar areas with probable investigation angles. We also aim to provide clinicians with adjustments in therapies, thus improving the realization of an effective treatment under COVID-19.

## 2. Methods

### 2.1. Search Strategy

In this article, we use a systematic review methodology to summarize current academic results made by other research. The electronic database research was conducted in February 2021, and included Pubmed, Chinese Science Citation Database (CSCD), the Web of Science Core Collection, Russian Science Citation Index, KCI-Korean Journal Database, Russian Science Citation Index, SciELO Citation Index, and Medline. For the database search, we use the following keywords while searching titles and abstracts: OCD (OCD, obsessive*, compulsive*) and COVID-19 (Coronavirus Disease 2019, SARS-CoV-2). We found 48 results on Pubmed, 2 results on Medline, and 2 results on the Web of Science Core Collection. Relevant articles, for instance, on the introduction of cognitive behavioral therapy (CBT) and exposure and response prevention (ERP), were found on both Pubmed and the Web of Science. All articles were written in English. The method of meta-analysis was not adopted for the heterogeneity in the included aspects of OCD.

### 2.2. Inclusion and Exclusion Criteria 

As is shown in Figure 1, firstly, we excluded 2 repeated articles since the same articles were found on different databases. Afterwards, we primarily excluded articles if they: (1) focused on diversified mental diseases besides OCD (*n* = 12); (2) investigated other diseases and clinical measures, simply using OCD as an example or evidence (*n* = 6). We obtained 31 articles after filtration. Among them, 2 were published by Chinese authors; 3 articles are from *Psychiatry Research*; 4 are from *Journal of Obsessive Compulsive and Related Disorders*. Eric A. Storch, from Department of Psychiatry and Behavioral Sciences, Baylor College of Medicine, USA, published 3 articles, which mainly focus on OCD therapies under COVID-19. Dean McKay, from Fordham University, USA, published 4 articles on related subjects. In order to choose those that were suitable for further discussion, we did a further filtration. We excluded articles if they: (1) particularly focused on people living in a restricted region (e.g., a province in Canada) (*n* = 4, since the sample quantity is small and limited, its result tends not to be prevalent); (2) were guidelines to clinicians (*n* = 2); (3) were reviews (*n* = 3); (4) only provided case reports, or were not involved in scientific research (*n* = 7); (5) used the pandemic as a background instead of investigating the connection between the pandemic and OCD (*n* = 3); (6) were an evaluation of a new scale (*n* = 1).

## 3. Results

In total, 52 records were identified, of which 2 were excluded for duplication and 37 were excluded according to the inclusion and exclusion criteria. A total of 13 articles were included in the review [7,8,9,10,11,12,13,14,15,16,17,18,19]. Two focused on adolescents and children [10,14]; ten focused on symptoms exacerbation [8,9,10,11,14,16,17,18,19]; one focused on treatments [12]; one focused on clinicians [13]; two focused solely on contamination fear [9,19]. Data resources of eleven research papers came from questionnaires. One research paper used clinical data only [12], one research paper also used clinical data [19]. Eleven adapted different scales to measure the severity of an individual’s OCD or depression. Nine used cross-sectional studies, one study also used panel data [13], and two studies were solely based on panel data [17,18]. Though preliminary, contamination fear becomes a focus on symptoms variation under COVID-19, since research on contamination fear is far more uncommon than those on other symptoms and obsessive-compulsive and related diseases (OCRDs). Symptoms may be prejudged by the first impression since an epidemic is closely related to sanitary conditions. Table 1 shows the main information of the 15 analyzed articles.

### 3.1. Studies on OCD Symptoms

Cox et al. [7] studied whether insomnia patients are vulnerable to OCD during the pandemic, and whether hoarding, checking, washing, etc. have been exacerbated under COVID-19. Their sample size was 379 insomnia- and anxiety-related disorder patients. OCD and depressive symptoms before coronavirus outbreaks contribute significantly to models that predict OCD symptoms after coronavirus outbreaks. Then, they added the symptoms of insomnia before the coronavirus outbreak to the model, which showed a 1% increase in the variation of previous OCD symptoms, indicating a significant R2 change (*p* = 0.02). Insomnia symptoms before a coronavirus outbreak predicted a slight increase in OCD symptoms after a coronavirus outbreak (B = 0.17, β = 0.10, *p* = 0.02), and depression symptoms before a coronavirus outbreak did not significantly predict OCD symptoms after a coronavirus outbreak. For specific symptoms, such as checking, neutralizing, ordering, and obsession, symptoms have not worsened. Hoarding and washing has worsened to different extents. Generally, OCD has worsened under COVID-19. 

Jelinek et al. [9] evaluated the changes on OCD symptoms during the quarantine and compared the distinguishment between contamination symptoms and remission state before and during the quarantine. The majority of participants (71.8%) reported the exacerbation of their OCD symptoms. The total Obsessive-Compulsive Inventory-Revised score was positively associated with small to middleweight changes in the general severity of OCD (R = 0.268, *p* ≤ 0.001), obsessive-compulsive disorder (r = 0.270, *p* < 0.001), mandatory factors (r = 0.304, *p* < 0.001), and circumvention (r = 0.208, *p* ≤ Measures). Participants associated increased mobility and reduced interpersonal conflict primarily. However, a small number of patients experienced symptom remission (6.5%) and 21.7% showed no symptomatic change. Their findings suggest that the majority of participants with OCD were negatively affected by the COVID-19 pandemic, and the negative effects were more pronounced in washers than in non-washers. This partly conflicts with Chakraborty et al. [19]. The latter points out that a small proportion of patients suffers symptom exacerbation under COVID-19.

Khosravani et al. [11] aimed to compare specific OCD symptom dimensions and symptom severity among a group of OCD patients before and during COVID-19. They used the COVID-19 Stress Scale (CSS) and Y-BOCS scales to measure patients’ situation. Their findings support the idea that the increase in symptoms and overall severity in people with OCD may be primarily due to stress caused by the current pandemic. In particular, stress responses associated with danger, pollution, traumatic stress, forced testing, and socioeconomic concerns predict worsening symptoms across multiple symptom dimensions. It highlights some of the potential mechanisms by which certain symptoms of OCD may be exacerbated by the current epidemic. Their finding is that the effects of COVID-19 are not limited to increased fear of pollution, but also involve other symptom levels, including liability for damage, unacceptable thoughts, and symmetry.

Storch et al. [17] investigated clinician cognition regarding the effect of the COVID-19 pandemic on patients with OCD receiving exposure and ERP prior to and during the pandemic. Their results relied on individual reports from patients instead of prospectively addressing patients. They found out that COVID-19 has been associated with ineffectiveness of ERP treatments in the majority of patients at the beginning of COVID-19. Their results estimated that 38% of their patients had symptoms worsen during the pandemic and 47% of them had symptoms remain the same. As the result, the majority of OCD patients’ symptoms did not increase. This conflicts with several studies [1,2,3,5].

Wheaton et al. [18] studied the relationship between emotion contagion and mental health symptoms during the COVID-19 pandemic. They invented the Emotion Contagion Scale to show patients’ reaction towards the pandemic. Their results indicated an approximately 14% of the variance in concerns about COVID-19 (R2 = 0.14, *p* < 0.001). Tests of individual regression coefficients showed that the Emotion Contagion Scale was a significant individual predictor of COVID-19 concern (b = 0.14 (SE = 0.02), *p* < 0.001) and daily consumption of COVID-19-related articles (b = 0.83 (SE = 0.20), *p* < 0.001). Significant predictors do not include daily consumption of social media (b = 0.17 (SE = 0.19), *p* = 0.37) and participant sex (b = 0.50 (SE = 0.63), *p* = 0.43). Their finding was that emotion contagion is a factor that leads to OCD symptom exacerbation. 

Nissen et al. [14] examined how adolescents with OCD have reacted towards the COVID-19 crisis. Tanir et al. [10] investigated the effects of COVID-19 pandemic and related self-quarantine on symptom profile, severity, and exacerbation of OCD symptoms among adolescents. Seçer et al. [16] investigated the relationship between fear of COVID-19 and OCD, conducting the research in a Turkish sample of 598 adolescents, whose results show that the effect of COVID-19 fear on OCD has been mediated by emotional reactivity, experiential avoidance, and depression anxiety. They are both pieces of research on adolescents and find that young subjects have been vulnerable to OCD under the pandemic, indicating that aggravation of OCD severity in young people is alarming.

Davide et al. [8] and Khosravani et al. [11] both measured symptom exacerbation by Y-BOCS scores. Davide et al. used a cross-sectional study while Khosravani et al. used a panel study. Khosravani et al. [11] also discussed the exacerbation of different symptoms. They both indicated that OCD has worsened significantly under COVID-19. 

Chakraborty et al. [19] aimed to evaluate the effect of COVID-19 on OCD patients, particularly those who have prior contamination fear. They made phone interviews with 84 patients who had prior contamination fear. The Yale–Brown Obsessive-Compulsive Scale (Y-BOCS) was used and researchers compared their scores with former ones. They found out that 48.8% patients had the Y-BOCS score stay the same or even decrease, 39.3% had a <5% increase, 6% had a 5–10% increase, 3.6% had a 10–25% increase, and 2.4% had a >25% increase. They did not find any increase in obsessive and compulsive symptoms in patients with contamination fear and washing before the pandemic. Only a very small proportion of patients (6%) have reported symptoms exacerbation, which is contrary to some studies ([7,8,9,11]), but is partly in line with [17]. Their conclusion is that handwashing promotion does not aggravate the washing compulsion of patients. Similarly, the fear of infection with the pandemic does not exacerbate contamination fear.

### 3.2. Studies on OCD Treatments

Kuckertz et al. [12] evaluated the effectiveness of traditional OCD therapies under COVID-19. Patients with OCD varied in the ways that they suffer from or deal with the pandemic. For certain patients, COVID-19 actually offers them opportunities or make them more motivated to engage in OCD treatment. Meanwhile, some patients did suffer from exacerbation of symptoms that are caused by COVID-19 or change their treatment plan due to the quarantine or other restrictions. Therefore, patients may not on average worry less about infection during the quarantine. This article points out that the effect of COVID-19 is a double-edged sword to OCD patients. Instead of prevalent exacerbation of symptoms, some patients experience an improvement of their mental health. However, this research does not point out the exact proportion of OCD remission.

Mckay et al. [13] investigated clinicians’ opinions on ERP under COVID-19. Their main models—the cold subscale of chills, the percentage of OCD cases, and the Intolerance of Uncertainty Scale Short Form anxiety subscale—were significant in predicting TBES (F(5127) = 21.35, *p* < 0.001, R2 (after adjustment) = 0.43). Similarly, the Intolerance of Uncertainty Scale Short Form apathy, percentage of OCD cases, and subscale of avoidance also had significance in predicting TBES (F(5,127) = 21.24, *p* < 0.001, R2 (after adjustment) = 0.43). As is shown by the research, clinicians’ caseload and a BIS activator predicted their concerns about exposure. The research suggests that personal clinicians’ concern about infection risks as they conduct treatment plans for OCD also matters while training clinicians. This study points out challenges for clinicians and discusses clinician approaches to OCD treatment under COVID-19.

## 4. Discussion

### 4.1. OCD Symptoms

Articles illustrate the exacerbation of OCD from different aspects. They discuss scores, symptom occurrence, remission rate, prevalence rate, and different age groups. The heterogeneous OCD symptoms can be summarized as eight classes [20]. The core symptoms of OCD are obsessions and compulsions [21]. Among those various symptoms, fear of contamination is the most prevalent [22], which is reported as increasing in severity by articles included and not included in this review [8]. Since washing hands thoroughly is recommended in order to prevent infection, this OCD symptom is the most frequently investigated [7,8,9,10,11,19]. Two in thirteen articles of research solely focus on contamination fear [9,19], and three refer to this symptom, all indicating that this symptom worsens under COVID-19. Some other research shows no difference before and under COVID-19 in contamination fear in certain regions, or if patients’ syndromes were formed before the pandemic [23]. Hoarding, which is characterized as a separated group of OCRD, is reported as aggravated. There are conflicts between Cox et al. [7] and Khosravani et al. [11] in several aspects. Cox et al. [7] suggest that there is not prevalent exacerbation in various OCD symptoms, since some of them, including checking, remain the same under COVID-19. Chakraborty et al. [19] and Storch et al. [17] hold the same opinion from another angle. The former finds that 51.2% of patients have symptom exacerbation, and the latter illustrates that 47% stay the same, which indicates that exacerbation is not prevalent. However, Khosravani et al. [11] illustrate that there is an extensive increase in various symptoms, and checking exacerbates under COVID-19. Therefore, there are conflicts between studies, but exacerbation under the epidemic exists. Despite the differences between the two studies, symptoms of depression that developed before the pandemic significantly and uniquely predicted a small increase in hoarding after the outbreak [7]. As for other specific symptoms, harm, unacceptable thoughts, and symmetry are reported to be exacerbated under COVID-19 [11]. 

Four articles included in this review use the Y-BOCS score to estimate the severity of OCD and all find out that there is a rise in the Y-BOCS score [8,11,17,19]. OCD prevalence rate is higher than it was before COVID-19 [24]. Some researchers found out that the number of relapsing patients with OCD grows [10,25]. As for different age groups, three articles mainly focus on children and adolescents [10,14,16]. It is probable that additional symptoms may develop, or original symptoms may worsen among young subjects [10,14,15,26], and they also have a high rate of remission, so their clinicians should adjust therapies accordingly. Moreover, despite the fact that remission is reported in other studies, there is no comparison of the remission rate among different age groups. The comparison would provide a good chance to estimate the ability of different age groups coping with OCD. 

The reason why OCD worsens under COVID-19 is also discussed by articles. As for exterior factors, COVID-19 is considered as a source of anxiety which spreads via emotional contagion. Consumption of media spreads anxiety [12], and inter-reaction between psychological disorders complicates under the pandemic. People mainly use social media to obtain the latest news of the pandemic. Results reveal that people spend hours viewing news of COVID-19 [27]. Perhaps, the constant exposure to social media which send news reports and health tips emphasizing the importance of self-sanitation, which may result in greater concern of being infected, becomes stimuli of OCD symptoms. For those whose symptoms are caused by interpersonal actions, their symptoms may alleviate temporarily. As for interior factors, it is also revealed that patients with insomnia are more likely to suffer from OCD under COVID-19 [7]. Other articles suggest that symptoms that refer to danger and contamination and remission status are important reasons for OCD exacerbation [8,28,29]. Thus, various factors should be included and adjustments should be made according to the patients’ personal situations.

### 4.2. OCD Treatments

The differences between OCD treatments before and during the pandemic can be concluded to four aspects: (1) there is a prevalent rise in anxiety in the social, economic, and political environment. (2) It is likely that hospitals and other treatment centers will be shut down for several months, reducing the efficiency of the communication between clinicians and patients. (3) Patients’ living environments are changed, as well as their social interaction ways and interpersonal relations. (4) A temporary absence from work can have an effect on patients’ mental situations. Pharmacological and psychological interventions for the management of OCD are wielded. Under COVID-19, as a result of impact on global economy and the adaptation of quarantine measures, variation in both methods occur. Articles on this aspect are relatively low in quantity. Among the selected articles, only one focuses on treatment methods.

ERP and CBT are the most prevalent ways to treat OCD, and are considered a gold treatment [30]. However, ERP courses’ effects on OCD treatments under COVID-19 are limited, and the mechanism behind them remains unclear [12,17]. As a key exterior factor, it is particularly worrying that patients’ living environments is transforming under COVID-19, causing an impediment to their treatments. For instance, support from family or individuals are indispensable for OCD therapies. Family members and caregivers of OCD patients are also at a greater risk of developing stress-related illnesses and may need additional support due to the worsening of the patient’s symptoms [31]. It can be concluded that OCD patients’ participation in therapies is affected, since anxiety is growing in their living environment. We assume that an increase in depression severity interferes with ERP, for it is suggested that aggravation of fear and depression may cause obstruction in OCD treatments [32]. 

Pharmacological treatments rely on medicine supply. The selective serotonin reuptake inhibitors (SSRIs) are the main choices of pharmacological treatment. Though solely listed as a control variable, there have already been reports that patients have stopped taking their medicines due to unavailability at the nearby drug stores [13]. Regulatory restrictions and virus-related manufacturing problems are disrupting global drug supply chains. Despite the fact that there are challenges to the reserve medical supplies in China [33], no reports claimed a rise in domestic SSRIs prices. However, the inevitable economic recession under COVID-19 may add a financial burden to certain patients or curtail medical supply channels, and domestic SSRIs may be more used than foreign ones. Moreover, it is concerning whether patients tend to increase their amount of medicine under COVID-19.

### 4.3. Implications for OCD Investigation, Therapies and Health Policy

It is undoubted that the epidemic offers a valuable chance to study OCD symptoms and recession. We assume that a complex interaction mechanism may lead to deterioration in some of these symptoms, while, as one’s living environment is considered safer or relatively different, the other symptoms may be remitted temporarily. For instance, the NIMH Global Obsessive Compulsive Scale (NIMH-GOCS) score increased in people with financial stress or unemployment, reflecting a deterioration in the severity of OCD. In contrast, patients without a financial burden showed relatively stable NIMH-GOCS [17]. It is also not hard to imagine that, for instance, a person with the fear of harming others may convert to contamination fear. More research is needed to understand more interactions between social factors that lead to the exacerbation of OCD symptoms. There is incongruity between studies, probably as a result of regional differences, of the unclear definitions of prior symptoms and of those that were newly formed during the pandemic, or of the limitation of sample capacity.

Both interior and exterior factors hinder OCD therapies. There are a number of concerns referred to this position [34]. We assume that the main obstacle for treatment effectiveness is the inefficiency of psychotherapy, since patients and their clinicians’ interaction is limited. As is illustrated, the improvement of the effectiveness of OCD treatment may need the cooperation between clinicians and patients [13]. Clinicians ought to have a proper attitude towards the pandemic and infection. They may search for effective ways and take advantage of current forms, for instance, telemedicine and smartphone interventions [35], to make effective communication with their patients, and adjust therapies simultaneously, taking advantage of remote assistance [36]. Nonetheless, face-to-face meetings are always recommended for their better therapeutic effects. They may also discuss their treatment procedures with patients explicitly. While these proposed adjustments attempt to take note of the current pandemic and CDC guidelines, it is important to note that deviations from established patterns can have potentially negative consequences, even if they are temporary. There is also a need to enhance the prevention of relapse during social restrictions as well as to develop other strategies [37], since there is research indicating that the relapsing rate rose during the pandemic [38]. Clinicians should not only treat patients that are still suffering from OCD, but also focus on the prevention of relapsing OCD [8]. Moreover, since psychiatric medicine is generally put on a longer-term administration and has relatively more doses than depression [39], to guarantee supplies for such medicines is indispensable under the pandemic, especially when hospitals or pharmacies are closed during the quarantine. Patients should be at least informed about their access to their drugs. Policies should be introduced to prevent shortages and price rising in peculiar medicine.

## 5. Conclusions

OCD exacerbation under COVID-19 is certain, though the extent of different symptoms varies. Although articles were published continuously on the impact of COVID-19 on OCD patients, researchers mainly focus on the exacerbation of contamination fear and the effectiveness of current therapies. The severity of the effect of COVID-19 is not estimated adequately. It is notable that the deterioration of OCD symptoms may cause difficulty to OCD treatments, and vice versa. COVID-19 interferes with the current CBT/ERP therapies, and adjustments towards treatments should be made. Among the 13 articles included, only one focuses on OCD treatments, indicating that treatment research is inadequate. Further research is needed on comprehensive learning about changes of patients’ living areas, people around them, their habits, etc., instead of solely putting scattered questions in a questionnaire. We also appeal for research on various symptoms and on comparison among different age groups, for the reason that COVID-19 may present a good chance to investigate connections between different syndromes. In all events, OCD investigation and treatment should progress under COVID-19.

## Figures and Tables

**Figure 1 ijerph-18-07420-f001:**
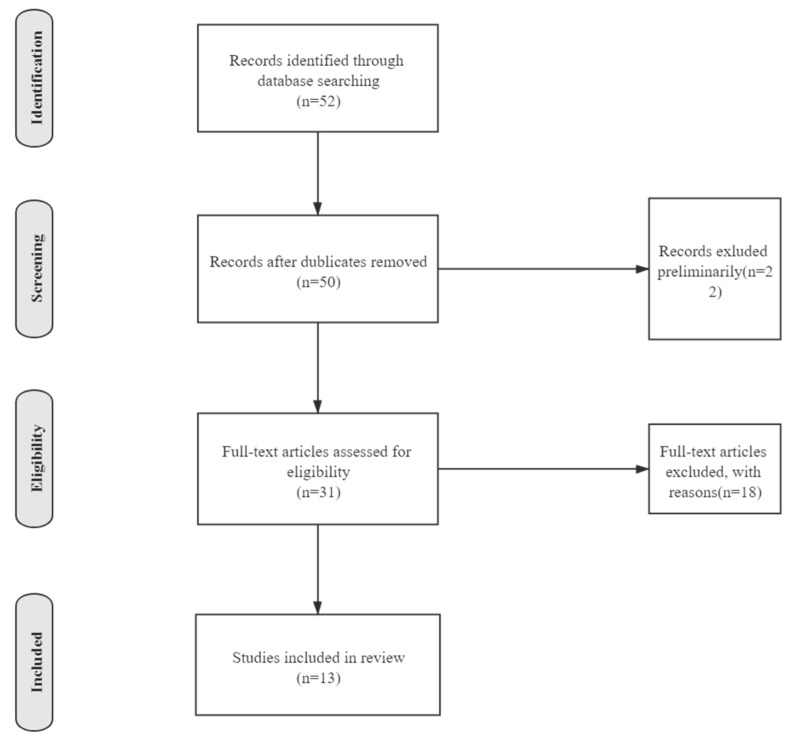
Flow diagram for the process of searching existing literature.

**Table 1 ijerph-18-07420-t001:** Data extraction of included studies.

Author and Year	Purpose	Data Sources	Study Design/Sample Size	Control Variables	Standards	Findings
[7] Cox et al., 2020	To determine whether insomnia patients are vulnerable to OCD during the pandemic, and whether peculiar symptoms exacerbates under COVID-19	Questionnaire data	Cross-sectional and panel/*n* = 379, Slight distinction between different researches	Living state, symptom	Changes in OCIR, ISI, DASS scores	Checking and some symptoms do notincrease. Hoarding and washing exacerbates to some extent.
[8] Davide et al., 2020	To evaluate the changes on OCD symptoms and investigated the effects of contamination symptoms and remission state	Questionnaire data	Cross-sectional/*n* = 30	Sex, remission status, social behavior	Increase or decrease in Y-BOCS score	During the quarantine, OCD worsens significantly, particularly amongst the patients withcontamination symptoms
[9] Jelinek et al., 2020	To determine whether OCD worsens more extraordinarily in those who are washers than those non-washers	Questionnaire data	Cross-sectional/*n* = 394	Washer and non-washer, symptom	Changes in OCIR scores	The negative effects of COVID-19 were more pronounced in washers than in non-washers.
[10] Tanir et al., 2020	To investigate the effects of home confinement exacerbation of OCD symptoms among young subjects.	Questionnaire data	Cross-sectional/*n* = 61	Age, sex, family income, treatment status, information sources, daily preoccupation current OCD diagnosis in parent(s), duration of parents’ education, COVID-19 diagnosis in someone familiar	Changes in CY-BOCS and CGI-S scores	Young subjects with OCD may develop additional symptoms and worsen already existing symptoms of OCD during COVID-19 pandemic.
[11] Khosravani et al., 2020	To compare a group of patients with OCD before and during COVID-19 on specific obsessive-compulsive symptom dimensions and symptom severity	Questionnaire data	Panel/*n* = 270	Symptom	Increase or decrease in Y-BOCS, DOCS, CSS score	The effect of COVID-19 occursacross various symptoms, including responsibility for harm, unacceptable thoughts, and symmetry.
[12] Kuckertz et al., 2020	To evaluate the effectiveness of tradition OCD therapies under COVID-19	Clinical data	Panel/*n* = 6	N/A	N/A	Effective OCD treatment can and should continue despite COVID-19.
[13] Mckay et al., 2020	To investigated the attitudes of mental health practitioners around exposure treatment during the COVID-19 pandemic	Questionnaire data	Cross-sectional/*n* = 139	Age, practising year, sex, ethnicity	Changes in CHILL, IUS-SF and TBES scores	Providers’ OCD caseload and a proposed indicator of BIS activation (coldness) significantly predicted their beliefs about exposure.
[14] Nissen et al., 2020	To examine how children/adolescents with OCD react towards COVID-19 crisis	Questionnaire data	Cross-sectional/*n* = 102	Age, gender, therapy, symptom	N/A	The study points towards an influence of the OCD phenotype, baseline insight suggesting a continued vulnerability.
[15] Rosa-Alcázar et al., 2021	To investigate coping strategies in OCD patients during COVID-19 lockdown	Questionnaire data	Cross-sectional/*n* = 122 (OCD patients), *n* = 115 (healthy people)	Age, sex, educational level, marital status, comorbidity	Results of COPE-28, HADS	Comorbidity affected the greater use of inappropriate strategies. Depression levels were related to the use of less adaptive strategies.
[16] Seçer et al., 2020	To investigate the relationship between fear of COVID-19 and OCD	Questionnaire data	Cross-sectional/*n* = 598	Age	Results of OCICV, Experiential Avoidance Questionnaire, DASC and ERS	The effect of COVID-19 fear on OCD is mediated by emotional reactivity, experiential avoidance and depression- anxiety.
[17] Storch et al., 2020	To investigate clinician perceptions regarding the effect of the COVID-19 pandemic on patients with OCD	Questionnaire data	Cross-sectional/*n* = 102	Sex, ethnicity, occupation, age	Changes in Y-BOCS and NIMH-GOCS scores	COVID-19 was associated with attenuation of ERP progress from expected rates.
[18] Wheaton et al., 2020	To study the relationship between emotion contagion and mental healthsymptoms during the COVID-19 pandemic.	Questionnaire data	Cross-sectional/*n* = 603	Sex, age, ethnicity	Results of CTS, ECS, OCIR, DASS-21	Greater susceptibility to emotion contagion was associated with greater concern about the spread of COVID-19 and OCD symptoms.
[19] Chakraborty et al., 2020	To determine whether washing compulsion and contamination fear aggravates under COVID-19	Clinical and questionnaire data	Cross-sectional/*n* = 104	Sex, religion, residence, occupation	Increase or decrease in Y-BOCS score	Handwashing protocol does not aggravate the washing compulsion of patients. The fear of infection with COVID-19 does not increase their fear of contamination.

ISI: Insomnia Severity Index. DASS: Depression, Anxiety, and Stress Scales-Short Form. CY-BOCS: Children’s Yale-Brown Obsessive Compulsive. CGI-S: Clinical Global Impression- Severity. HADS: Hospital Anxiety and Depression Scale. DOCS: Dimensional Obsessive Ccompulsive Scale. IUS-SF: Intolerance ofuncertainty scale-short form. CHILL: The chills scale. TBES: Therapist Beliefs about Exposure Scale. COPE-28: Spanish Version of the Brief COPE. OCICV: Obsessive Compulsive Inventory-Child Version. DASC: Depression and Anxiety Scale for Children. ERS: Emotional Reactivity Scale. CTS: COVID-Threat Scale. ECS: Emotion Contagion Scale. DASS-21: Depression, Anxiety and Stress Scale.

## Data Availability

No data, models, or code were generated or used during the study.

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
