# Peer review of "Variation in Obsessive-Compulsive Disorder Symptoms and Treatments: A Side Effect of COVID-19"

_ijerph, 2021, doi:10.3390/ijerph18147420_

Round 1

Reviewer 1 Report

Thank you for your responses, comments and changes. It is a shame that your university does not allow you access to PsycInfo but it is unfortunately more and more frequent. All changes to my comments are satisfactory. There are still some shape changes like an uppercase line 277 but I'll let the editors take over!

Author Response

Dear reiviewer 1:

    Thank you for your comment. We have adjusted the table to make it concise and rectify grammar mistakes in the article. 

Reviewer 2 Report

The authors have taken into account the comments from the reviewers and the manuscript has been improved. The table requested by both reviewers is interesting but too heavy in its current form. I suggest that the authors summarize the "purpose" and "findings" columns of the table with relevant key words (and not lengthy sentences). They should also provide a legend for the table, including the abbreviation list, and pay attention to the layout, spaces, ponctuation, etc. Finally, they should refer to the table in the main body of the text when appropriate. Thus, I suggest that they provide minor revisions to the table.

Minor comment: Please check the first mention of the abbreviations in the text. For example, CBT is defined only at the second appearance. Also consider not to use an abbreviation when it is utlized only twice (e.g. IUS-SF, SSRI...).

Author Response

Dear reviewer 2:

   Thank you for your comment. We agree with your suggestion and have made improvement. We made our table succinct and added legends for each abbreviation. After consulting other reviews in our references, we choose to shorten our sentences in the table since key words are not sufficient enough to summarize the main ideas, and we pay attention to the appearance of the table. We put a sentence to indicate the existence of the table in line 101. Finally, the places of the first mention of CBT and ERP are moved, and some abbreviations used only 1 time are removed.

Best regards,

authors

Reviewer 3 Report

The authors’ revised the manuscript according to reviewer comments. However, there are still uncertain points.

  • The authors stated a summary table of the articles at the end of the article. The reviewer checked the revised manuscript and the files for the reviewer thoroughly, but could not find it.
  • ‘We have several conclusion that is multivariate. We did some adjustments in the article, such as the location of the ‘result’ paragraph. We hope this changes can help readers better understand our views’. The authors are dodging the point. The reviewer asked comprehensive analyses, not only citing multivariate analysis paper.

Author Response

Dear reviewer 3:

Thank you for your comment. We have carefully taken your comment into consideration. You can find the table at the end of the article as pictures, and the PDF version will be submitted seperatedly. We improved our article according to your comment and made comprehensive analysis, as you can find in the revised version. Thank you again for your help.

Best regards,

authors

This manuscript is a resubmission of an earlier submission. The following is a list of the peer review reports and author responses from that submission.

Round 1

Reviewer 1 Report

The article is very interesting and really corresponds to a contemporary clinical concern. However, this could be easily improved by modifying grammar and punctuation. Words, periods, capitals, spaces are missing, as well as some unfinished sentences. In addition, a few points could be improved.
The summary presents the main points of the study by summarizing the methodology and the points discussed according to the results. It might as well include some more specific results to answer the main question of the article. The Key Message is sufficient, but it should end with a period.
The introduction helps to situate the context of the study and its current interest during the pandemic.
At the end of the introduction, the abbreviation OCRD appears, we assume it stands for Obssessional-Compulsive Related Disorder but it would be helpful for readers to recall its meaning at least at the first occurrence in the text. The same goes for CBT and ERP in the "search strategy" paragraph; then for OCI-R, CSS, Y-BOCS, ECS, IUS-SF, TBE, BIS, NIMH-GOCS and CDC in results and discussion.
Regarding the methodology, it is a great pity not to have used the "Psycinfo" database for bibliographic research because it is one of the most indexed databases for psychiatry, psychology and psychopathology. The authors must therefore have missed a number of studies. Could you check that out?
In the methodology, why have the articles referring to patients living in small regions deleted? What is a small region? Is there any scientific knowledge on the difference between large and small region for patients with OCD related or unrelated to the pandemic?
The "Results" paragraph should be paragraph 3. A summary table of the main studies presented would be welcome. This would make it possible to lighten the drafting and thus avoid certain repetitions that are a little too present between the results and the discussion, even if it means reducing the length of the article.
There are missing spaces between commas and there is too much white space on line 96. Likewise, this sentence is not understood: if Cox et al. study insomniac patients at risk for OCD how come their sample is made up of 379 OCD patients. Is it not rather the reverse?
Line 119, there is mention of contradictory results with another study (Chakraborty et al.). It would be interesting to present the points of contradiction to help the reader become aware of them. Perhaps, quite simply, by bringing the paragraph referred to. Line 139, a space is missing between the parenthesis and "study".
With regard to adolescents, why are some studies described and others not? The sentence in line 170 is not finished.
There is no paragraph for study limitations. These are scattered all over the discussion. Is it possible to put them together in a single paragraph. Likewise, the discussion and the results should be homogenized because the entire beginning of the discussion (lines 197-227) are only additional results or even repetitions.

Reviewer 2 Report

The systematic review article is very interesting to read. However it suffers from a lack of structure and interpretation regarding the description of the results of the articles considered in the review.  I suggest the authors to provide a clearer structure to describe the different papers enclosed, better interpret the papers and the discrepant data. A table summarizing the studies considered in the review (characteristics of the individuals, main findings) would help the authors and the readers to understand the main messages.

The English scientific writing could be improved. Please define the abbreviations before first mention. 

Reviewer 3 Report

No display items except inclusion/exclusion procedure provided. The authors cited couple of papers, but readers have to read each paper to refer results. As of now, the paper consists of enumeration of published papers of OCD and COVID-19. Multivariate analyses which support the readers’ conclusion should be performed.